# A Qualitative Study Exploring Professional Perspectives of a Challenging Rehabilitation Environment for Geriatric Rehabilitation

**DOI:** 10.3390/jcm12031231

**Published:** 2023-02-03

**Authors:** Lian M. J. Tijsen, Els W. C. Derksen, Wilco P. Achterberg, Bianca I. Buijck

**Affiliations:** 1LUMC, Leiden University Medical Center, Department of Public Health and Primary Care, Postzone V0-P, Postbus 9600, 2300 RC Leiden, The Netherlands; 2Oktober, 5531 LJ Bladel, The Netherlands; 3De Zorgboog, 5760 AA Bakel, The Netherlands; 4Radboud University Medical Center, Department of Primary and Community Care, 6500 HB Nijmegen, The Netherlands

**Keywords:** geriatric rehabilitation, challenging rehabilitation environment, post-acute care, care process

## Abstract

There is a trend towards the formalization of the rehabilitation process for older rehabilitants in a Challenging Rehabilitation Environment (CRE). This concept involves the comprehensive organization of care, support, and environment on rehabilitation wards. So far, literature on the principles of the CRE is scarce. This study aims to explore the perspectives regarding the CRE of healthcare professionals through a qualitative study. Therefore, between 2018 and 2020, six international and 69 Dutch professionals were interviewed in focus groups, and 180 professionals attended workshops on two Dutch congresses. Data were thematically analyzed using ATLAS.ti. Seven themes emerged regarding the rehabilitation processes: (1) rehabilitant (attention for cognitive functioning and resilience); (2) goals (setting personal goals); (3) exercise (increasing exercise intensity); (4) daily schedule (following the daily rhythm); (5) involving the client system (involving informal caregivers); (6) nutrition (influences rehabilitation capability); and (7) technology (makes rehabilitation more safe and challenging). Regarding organizational aspects, four main themes were identified: (1) environmental aspects (encourages exercises); (2) staff aspects (interdisciplinary team); (3) organizational aspects (implementing CRE requires a shared vision); and (4) factors outside the ward (a well-prepared discharge process). To offer effective rehabilitation, all elements of the CRE should be applied. To improve the CRE, specific interventions need to be developed and implemented. Consequently, the effectiveness and efficiency of the CRE need to be measured with validated tools.

## 1. Introduction

Rehabilitation involves the identification of a person’s problems, challenges, and needs. This leads to defining rehabilitation goals and subsequent interventions offered by a multidisciplinary team [1]. Persons undergoing rehabilitation are trying to adapt and self-manage their current condition, and, in line with the ideas of Huber et al. on positive health, the term “rehabilitants” is therefore more appropriate than the term “patients” [2].

A specific form of rehabilitation is geriatric rehabilitation (GR), which has recently been defined as ‘a multidimensional approach of diagnostic and therapeutic interventions, the purpose of which is to optimize functional capacity, promote activity and preserve functional reserve and social participation in older people with disabling impairments’ [3]. Rehabilitation for older people is even more challenging than for younger persons. With the ageing of the population, the demand for GR in Europe has increased [4]. In 2019, 53,320 rehabilitants in the Netherlands were referred to GR [5]. After hospitalization on an acute geriatric ward in Italy, 11% of patients aged ≥75 years were referred to a rehabilitation unit [6]. Common reasons for hospitalization in older persons are cardiac events, infections, fall-related injuries, stroke, cancer, or medical/surgical interventions [7].

In the Netherlands, there is a trend towards having the rehabilitation process take place within the context of a challenging rehabilitation environment (CRE). A CRE is a widely accepted practice-based innovation in the Netherlands [8,9,10]. It is a unique concept which is positioned on the rehabilitation ward, covering all rehabilitation aspects that possibly influence rehabilitation outcomes. The concept involves the comprehensive organization of care and support by the rehabilitation team as well as the environment in which the rehabilitation takes place [8,11]. In comparison to regular rehabilitation (with mostly mono-disciplinary goals and interventions), in a CRE, the rehabilitation interventions are integrated into all aspects of the day and daily life, and the rehabilitation process is offered in an interprofessional way, with team and rehabilitant goals and interventions [12]. 

A review conducted by the authors initially identified seven main components for modelling CRE: (1) therapy time; (2) group training; (3) patient-regulated exercise; (4) family participation; (5) task-oriented training; (6) enriched environment; and (7) team dynamics [11].

Although internationally there is a growing interest in the principles of this relatively new concept, there is no official scientific-based definition of the CRE. This results in considerable differences between rehabilitation wards in the interpretation of rehabilitation in general and the execution of a CRE specifically. Because of these differences in interpretation, the seven mentioned components are not guaranteed to cover all aspects of the CRE, and the question emerges regarding which aspects are found to be relevant by stakeholders besides the seven aspects found in the literature.

To date, no evidence-based conceptualization of CRE has been developed, and empirical evidence for the added value of CRE for rehabilitants is lacking. The current study is part of the CREATE study (Challenging REhAbiliTation Environment) [13]. In this part of the CREATE study, we explore the perspectives of professionals regarding CRE.

## 2. Methods

### 2.1. Study Design

To explore the perspectives of professionals on the concept of the CRE, a qualitative study consisting of focus groups and workshops was performed between September 2018 and January 2020. The primary aim of qualitative research is to gain a better understanding of a phenomenon through the experiences of those involved [14]. As not all components relevant to CRE are identified, a qualitative study is indicated to gain a better understanding of the concept of the CRE. A waiver of consent was issued by the Medical Ethical Committee of the Leiden University Medical Center. This study did not apply to the Medical Research Involving Human Subjects Act (N19.024) [11,13]. 

We adhered to the consolidated criteria for reporting qualitative research (COREQ), which aim to improve the quality of reporting this type of research (see Appendix A) [15].

### 2.2. Recruitment of Participants

Participants were eligible if they had recent experience in the field of (geriatric) rehabilitation and were willing to sign an informed consent form after receiving verbal and written information on the study. 

During the recruitment phase, we aimed at obtaining input from different professional disciplines in the broad field of rehabilitation, e.g., medical doctors, physical therapists, occupational therapists, psychologists, speech and language therapists, and nurses, as well as researchers in the field of GR.

Recruitment for the focus groups was split into three target audiences: experts, (para)medics, and nursing staff. This subdivision was made to ensure that all participants were able to express their perspectives, regardless of educational or hierarchical issues.

At the start of the study, we composed a list of 31 Dutch national experts in the field of (geriatric) rehabilitation. These experts were invited by email to participate and were asked to supplement the list with names of people they regarded as experts. This resulted in a list of 38 people who were invited to participate in these focus groups.

Furthermore, an international focus group was established with non-Dutch members of the Special Interest Group Geriatric Rehabilitation (SIG GR) of the European Geriatric Medicine Society (EuGMS) [16]. Fourteen members were asked via email to participate in a focus group during the EuGMS congress 2019 in Krakow. The audience of the symposium of SIG GR at this congress was also invited to participate in this focus group.

Organizations affiliated with the six academic networks for elderly care in the Netherlands were approached by email with information about this study [17]. They were asked to participate in this study and to delegate professionals working in the rehabilitation field to attend the focus groups.

We aimed for data saturation, and after each focus group, the authors discussed whether any new topics had emerged. Inclusion stopped when all disciplines were represented in focus groups and no new topics emerged.

Additionally, the study group was asked to organize four workshops concerning the theme CRE during two Dutch national congresses in the field of rehabilitation. This opportunity was used to ask the participants of these congresses to provide input on topics relevant to CRE. Visitors of these congresses were able to register for these workshops, and they were informed that their input was used for the conceptualization of CRE. Participation in the workshops was voluntary, and no personal information was collected.

### 2.3. Focus Groups

The aim of the focus groups was to clarify the perspectives of the participants regarding CRE. The groups were chaired by E.D. or B.B., and L.T. took field notes during the group interviews. E.D. and B.B. are both female senior nursing researchers in the field of rehabilitation. Both are experienced in qualitative research and chairing discussion groups. Physical therapist L.T. is a female PhD student with formal training in interview techniques and qualitative research and has 10 years of experience in geriatric rehabilitation.

In preparation for these groups, L.T. developed a topic list (Appendix B) based on an earlier literature review on CRE [11]. This topic list was piloted with a group of researchers. The content of the topic list was determined in an iterative process and adapted based on the previous pilot and focus groups.

The focus groups for Dutch experts were held in a meeting center centrally located in the Netherlands. The international expert group was held at the congress location of EuGMS 2019, Krakow. The focus groups for the other professionals were conducted at meeting centers and rehabilitation wards spread across the Netherlands. All focus groups took place in meeting rooms, and only the participants and researchers were present.

Each focus group began with a brief introduction to the study and the topic of the focus groups, followed by the introduction of the individual participants. The participants were then asked to share their perspectives on CRE. The chair asked open-ended questions based on the topic list to keep the conversation going. To increase the internal validity, participants were also asked to share their perspectives on subjects that were not included in the topic list but which they considered important regarding CRE.

On average, the duration of the focus groups was 110 min, and they were audiotaped and transcribed verbatim by L.T. Transcripts were not returned to participants for comments, but at the end of every focus group, the chair presented a verbal summary and checked its accuracy with the participants.

### 2.4. Workshops

Four 50-min workshops were held at two Dutch national congresses in the field of rehabilitation. Each workshop started with a presentation by L.T. about the results of the review on CRE [11]. Participants were informed of the purpose of this qualitative study, and after the presentation, the participants split into groups of 8 persons on average. In these groups, they discussed one of four questions compiled by the researchers regarding CRE. L.T. and B.B. guided these discussion rounds, and participants were asked to summarize the results of their discussion on a flipchart and present them to the other participants. These flipcharts were digitized and used as an input for data analysis.

### 2.5. Data Analysis

Parallel to the data collection, we performed a thematic analysis to identify, analyze, and report patterns in the data [14,18]. For coding of the data, ATLAS.ti version 7.5 was used. 

L.T. familiarized herself with the data by reading and re-reading the transcripts, after which initial themes were identified using an open-coding approach. These initial themes were checked and coded by B.B. and E.D. to determine inter-rater agreement. Differences in the coding were discussed by L.T., E.D., and B.B. until an agreement was reached. Each initial theme was described in a memo. 

The identified initial themes were combined into main themes with associated sub-themes. The connections and contradictions between the initial themes were described per main theme and connections between main themes were described in categories. 

Each main theme was assessed for data saturation by checking whether no new data emerged in the focus groups or workshops. Subsequently, the research team discussed the main themes. After an agreement was reached, each main theme was thoroughly described, and relevant quotes were identified and translated into English.

## 3. Results

### 3.1. Participants

Between September 2018 and October 2019, a total of 13 focus groups were conducted: one international expert group (n = 6), three national Dutch expert groups (n = 17), three (para)medics groups (n = 24), and three groups with nursing staff (n = 28). Eleven of the invited Dutch experts and eight of the invited international experts were not able to participate. Reasons for not participating were time management concerns, not being present at EuGMS congress 2019, or having the impression of not having sufficient knowledge about the subject.

The workshops were held in November 2019 and January 2020. A total of 180 rehabilitation professionals participated in these workshops.

Characteristics of the participants are shown in Table 1.

### 3.2. Themes

Eleven main themes with associated sub-themes emerged from the data. The main themes can be divided into two categories, namely themes involving rehabilitation processes and themes involving organizational aspects. The subdivision of the themes within the two categories is described in Table 2 and Table 3. The two categories are described in the following paragraphs.

#### 3.2.1. Category 1: Themes Involving Rehabilitation Processes

This category consists of seven main themes: (1) rehabilitant; (2) goals; (3) exercise; (4) daily schedule; (5) involving client system; (6) nutrition; and (7) technology.

##### Theme 1.1: Rehabilitant

A CRE is suitable for all diagnosis groups, although the principles of CRE must be introduced to rehabilitants and their informal caregivers to stimulate self-reliance.

During the rehabilitation process, attention should be paid to potential cognitive problems, sometimes pre-existent or sometimes newly emerged. One nurse practitioner stated:


*Especially in CVA patients, depression is quite common and very often underestimated. ... But this has to be included, because if the mood is not right, there is a very negative impact on the rehabilitation process.*


As an elderly care physician said, it is important to take the rehabilitant’s resilience into consideration in their day program:


*But 1 time 24 minutes is not the same as 24 times 1 minute. And those 24 times 1 minute is what you want in a CRE. You can also spread patients with limited abilities over the day so that they can still continue in therapy, despite their limits.*


##### Theme 1.2: Goals

In GR, it is important to work on the rehabilitant’s own goals for motivation, self-reliance and independence. Sometimes, rehabilitants need guidance from professionals or informal caregivers to describe their goals. A manager with a background as a physical therapist explained:


*It starts with a good talk and actually motivating the rehabilitant. Everyone is motivated for something, but maybe not for your goals.*


Participants miss measurement instruments validated for the GR populations to measure the success of rehabilitation. As one rehabilitation physician stated:


*I want to advocate defined clinimetrics. To inform [rehabilitants] properly and measure treatment success.*


##### Theme 1.3: Exercise

To achieve the highest possible exercise intensity, training moments must be integrated into the daily routine. With task-oriented exercises, rehabilitants train meaningful tasks aimed at a participation level. A rehabilitation physician explained:


*So, the question is, how do you integrate exercise components in the daily routine. …. So, I think, this is really, let’s say, the big picture. That we have to change the climate of how we work with the people.*


Patient-regulated exercises can increase the exercise intensity and stimulate the rehabilitants’ independence. If group training is focused on the goals of a rehabilitant, it can increase the exercise intensity and stimulates contact with other rehabilitants. An elderly care physician said:


*I think group therapy can be very efficient. ... It may help when people practice in a group and you have peer support.*


##### Theme 1.4: Daily Schedule

Within a CRE, across the entire day, all activities should be focused on rehabilitation, and those activities should be stimulated.

Participants are in favor of working without strict planning to be able to respond to the rhythm of the rehabilitant and stimulate interdisciplinary cooperation. As a nurse mentioned:


*I would prefer to have one occupational therapist and one physiotherapist on the ward structurally. Who can just help out on the ward from morning to evening, and at the same time provide therapy.*


##### Theme 1.5: Involving Client System

The client system can be seen as fellow practitioners in the rehabilitation process, but staff must guard against overburdening the informal caregiver. One nurse and lecturer explained:


*I do think it’s important that the family caregiver has a place and is a natural part of the whole. I also think it’s very important that we are aware that, from the family caregiver’s perspective, there is no end to it.*


To be able to involve the informal caregivers in the rehabilitation, communication is a key aspect. As an elderly care physician and researcher said:


*What I also see a lot ... is that even for family caregivers it is often unclear what they should expect. What the approach will be and what the goal of the rehabilitation ward is. In addition to everything we have already said, I think that explaining and providing information is also an important part of the rehabilitation climate.*


##### Theme 1.6: Nutrition

The nutritional status of a rehabilitant partly determines their workload capacity. So, attention to a protein-rich diet with the most common products possible is important. One elderly care physician mentioned the importance of nutrition:


*Nutritional status is another one. Yes, it’s getting more attention now, but it has been underexposed for a very long time I think. And also the link with people sometimes just being too tired to eat properly. And I’m not even talking about the quality and how tasty it is, so to speak.*


##### Theme 1.7: Technology

Technology develops very fast and contributes to safe and challenging rehabilitation. Currently, eHealth is not often used in the GR, but as one nurse practitioner summarized:


*Well, it has a lot of potential, but the tricky thing is, there are so many applications. Remember you are dealing with elderly people who have difficulty with technology and you have to organize your whole care process in such a way that the technology takes this into account. So, to implement it properly, there are quite a few conditions to meet.*


#### 3.2.2. Category 2: Themes Involving Organizational Aspects

Four main themes belong to this category: (1) environmental aspects; (2) staff aspects; (3) organizational aspects; and (4) factors outside the ward.

##### Theme 2.1: Environmental Aspects

The environment on a rehabilitation ward must be safe and must invite rehabilitants to practice as much as possible. As one physical therapist said about the building aspects of a rehabilitation ward:


*It is an interaction of a warm environment that is very stimulating and invites to start doing the things required to be able to go home.*


It is important that the environment resembles the domestic situation as much as possible, and everyday equipment is used. The environment should stimulate rehabilitants to practice as much as possible, and rehabilitants must have access to exercise materials all day. This can be achieved by providing exercise opportunities in the corridors and possibilities to go outside. As one manager said:


*When you get to the point in the rehabilitation process that you are able to practice independently on the parallel bars, then you want to do that as often as possible, I’d think. I would like to have that nearby or be allowed to go there on my own to practice. Then I can imagine it being on the ward is convenient.*


##### Theme 2.2: Staff Aspects

A rehabilitation team must work in an interdisciplinary way, and the rehabilitant and their informal caregiver are seen as part of the team as well. As an elderly care physician explained:


*If you are referring to interdisciplinary working. That’s a core concept in rehabilitation. You have the specialist expertise in all fields, but you also have to know and be able to borrow from each other’s expertise a little bit.*


All employees must have an emphatic, motivating attitude and stimulate rehabilitants to practice throughout the day. As a rehabilitation physician stated:


*I think, particularly nursing staff having a rehabilitation focus, and so, encouraging for the patients to do everything possible they can, from the start. So, that may make a significant difference.*


As rehabilitants in the GR do not always fit in medical guidelines, staff must be able to work based on the ideas of evidence-based practice. As one physical therapist said:


*That is the problem with the application of such a guideline. For example, the guideline says it’s for stroke, but if someone also has Parkinson’s, or broke his hip last year, you cannot do certain tests. Because it’s obviously impossible.*


##### Theme 2.3: Organizational Aspects

Even though internationally organizational aspects differ and can therefore influence the rehabilitation process, the concept of CRE is suitable to get the most out of rehabilitation. Implementing a CRE requires a shared vision on rehabilitation and a balanced interdisciplinary team with sufficient time for the implementation. As a nurse practitioner said:


*I do think when you have that kind of project group, it does involve regular evaluation. Like, guys, how are the things we started going now? And do we need to adjust, fine-tune anything.*


CRE does not depend on the rehabilitation setting, as long as the name of the ward does not generate false expectations. It is also important that rehabilitants are not addressed as patients. As a nurse lecturer said:


*Calling someone patient or client, you emphasize what a person can’t do. If you say person, you avoid this label. It is still someone who tries to live his life in the best way possible.*


##### Theme 2.4: Factors Outside the Ward

The discharge process must be well prepared and supervised. Home visits allow rehabilitants to practice meaningful tasks in their own environment in preparation for their discharge. One manager with a background as physical therapist mentioned:


*It also helps to have people actually go home during rehabilitation. This provides so much information about how they actually function at home. A situation is always different at home.*


Although participants think it is a good idea to organize rehabilitation in the home situation as soon as possible, they also doubt whether it is better to keep rehabilitants on the rehabilitation ward for longer. As an elderly care physician said:


*The question is, if you have a ward with a very good rehabilitation climate, would you not want to admit patients there who, in terms of their care needs, could go home, but for whom the added value of the rehabilitation climate for the rehabilitation is such, that patients choose to be admitted to the department for rehabilitation.*


## 4. Discussion

This article is the first to describe the perspectives of healthcare professionals in rehabilitation, concerning CRE. A set of seven factors concerning rehabilitation processes and four factors concerning organizational aspects emerged from the qualitative data. The results of this study are (partially) in line with our review on CRE and confirm the importance of increasing therapeutic intensity, the importance of patient-regulated exercise, group training and task-oriented training in a CRE. Involving informal caregivers, providing a challenging environment for rehabilitation, and a cooperating, motivating team are also aspects of a CRE. These factors, therefore, constitute challenges for a rehabilitation team to work on [11].

Participants in the current study believe a CRE is suitable for all types of rehabilitants, but it has to be tailored to the resilience, goals, and cognition of the rehabilitant. This is in line with recent literature, which states that rehabilitation is suitable for persons with all kinds of diagnoses when it is tailored to the needs, goals, and wishes of the individual rehabilitant [19,20]. This confirms the relevance of tailoring the rehabilitation process to the individual rehabilitant, which is the main challenge for professionals in a CRE.

The results of this study indicate that it is important in a CRE to work on a rehabilitants’ own goals and to measure them with appropriate measurement instruments. Although a recent meta-analysis could not substantiate its added value, they do see goal setting as a part of shared decision-making and as a way to respect the preferences, values, and autonomy of rehabilitants [21]. A recent review endorsed the importance of personal, meaningful goals for rehabilitants and described the importance of involving rehabilitants in and informing them about the process of goal setting [22]. Although the added value of goal setting with the rehabilitant requires further research, both the literature and participants of this study consider it of interest in the rehabilitation process. Involving the rehabilitant in the goal setting in a CRE is therefore recommended.

Participants in this study feel that attention to the nutritional status of a rehabilitant is relevant and an optimal “food as usual” but protein-rich diet should be the goal. This is endorsed by recent literature that indicates that the nutritional status is significantly related to successful rehabilitation in older adults [23,24]. The results of this study and the literature indicate the importance of attention to the nutritional status of rehabilitants. Consequently, professionals in a CRE should measure the nutritional status and choose nutrition activities accordingly.

Although eHealth is currently not often used in GR, participants do see a lot of potential in eHealth for exercising, monitoring and safety during practice. Applications must be suitable for the target group. Systematic reviews show the benefits of eHealth for older persons in terms of increasing their physical activity, walking ability, and balance [25,26,27]. A recent review on eHealth in GR confirms the benefits of integrating eHealth in GR [28]. Although more research regarding eHealth in GR is necessary, the literature so far confirms the potential of eHealth for rehabilitants in GR. Therefore, eHealth may be an important aspect in CRE, and professionals should look for ways to apply eHealth in a functional way in a CRE.

A CRE can be negatively affected by organizational aspects such as funding, administrative tasks, and legal regulations, which can differ internationally. Participants think it is important that an organization has a shared vision on rehabilitation and there is sufficient time for implementing a CRE. 

Implementing a complex concept such as the CRE should be based on an understanding of the behaviors that need to change, the relevant decision-making processes, and the barriers and facilitators of change. Monitoring during and after the implementation is crucial [29]. The literature confirms the idea mentioned by participants in this study regarding the complexity and barriers to implementing a CRE: successful implementation of a CRE on a ward requires a strategy and sufficient time for the implementation process.

Participants stressed the importance of practicing meaningful tasks in a rehabilitant’s own environment and thus of rehabilitation in the home situation as quickly as possible. However, some are in doubt as to whether a longer stay on the rehabilitation ward is better if there is a good CRE. As the effectiveness and efficiency of CRE have not yet been studied, no statement can be made about the benefits of inpatient rehabilitation in a good CRE versus outpatient rehabilitation at one’s own home. This needs to be the subject of further research.

The 11 themes that were identified form a rather complex concept. In general, the rehabilitation process should be individually tailored and optimized to achieve all the goals of the rehabilitant. Currently, all principles of CRE are used internationally in GR. However, the rehabilitation ward may not work according to all of the themes that are important for a CRE. Therefore, new interventions should be implemented and adapted.

The strength of this study is the number of participants. We interviewed more than 200 individuals, and data saturation was reached. Secondly, all professionals participating in this study had experience in the field of rehabilitation. The occupation of the participants was taken into account in the composition of the focus groups. Therefore, hierarchical differences did not prevent participants from discussing their ideas, although it also limited the exchange of ideas between groups. In the workshops, participants were mixed in smaller groups regardless of occupation. The results of these workshops were in line with the results of the focus groups, meaning that occupation did not influence the results. The use of focus groups and workshops stimulated the exchange of ideas, which also resulted in new ideas. Participants were asked for subjects they thought were important for a CRE, even when not asked for by the researchers. In this way, it was ensured that all relevant topics were discussed and the internal validity of the study was increased.

A limitation of our study is that most participants are from the Netherlands and were somehow familiar with CRE ideas. This may limit the generalizability of the results to GR in other countries in which the concept of CRE is in its infancy. However, the topics discussed in the non-Dutch focus group at the 2019 EuGMS congress were in line with the results of the other focus groups. We, therefore, think that the identified themes are important for all rehabilitants in GR, regardless of the country in which they are rehabilitating. 

## 5. Conclusions and Implications

Based on this study, 11 themes were identified for modeling a CRE. Overall, it is important to tailor the rehabilitation process to the rehabilitant and to stimulate rehabilitants to optimize their rehabilitation. 

Since tailoring the rehabilitation process in a CRE to rehabilitants and their informal caregivers seems important, it is interesting to investigate whether these eleven themes are supported by the rehabilitants themselves and to find out if they consider other factors important for a CRE. According to the respondents, to offer effective rehabilitation, all elements of CRE should be applied, and specific interventions need to be developed and implemented. Consequently, the effectiveness and efficiency of CRE need to be studied with validated tools that are yet to be developed. In our ongoing research, we aim to develop those tools.

## Figures and Tables

**Table 1 jcm-12-01231-t001:** Characteristics of participants.

Expert Groups (n = 4, Participants = 23)		
Nationality	Netherlands	17
	Greece	1
	Germany	2
	Australia	2
	Poland	1
Occupation *	Elderly care physician	5
	Rehabilitation physician	4
	Physical therapist	11
	Lecturer	2
	Researcher	11
	Manager	8
	Nurse (practitioner)	2
**Focus Group (para) medical (n = 3, participants = 24)**		
Occupation	Elderly care physician	3
	Occupational therapist	5
	Nurse practitioner	3
	Physical therapist	8
	Psychologist	1
	Speech and language therapist	3
	Head of nursing staff	1
**Focus Group Nursing Staff (n = 3, participants = 28)**		
Occupation	Nurse	22
	Nurse assistant	4
	Nurse aid	2
**Workshops (n = 4, participants = 180 healthcare professionals, e.g., physical therapist, occupational therapist, elderly care physician, nurse)**		

* Participants indicated combinations of occupations.

**Table 2 jcm-12-01231-t002:** Themes involving rehabilitation processes.

Main Theme	Brief Description	Sub-Theme	Description
Rehabilitant	CRE is suitable for all diagnosis groups. Attention to rehabilitants’ cognitive functioning and resilience and stimulating the self-reliance of rehabilitants are necessary.	Characteristics	Rehabilitants undergo rehabilitation for different diagnoses, e.g., in the fields of neurology, orthopedics, and trauma. They often have multiple diagnoses and are already experiencing a functional decline in the home situation. They are often not familiar with using technologies to perform exercises. Traditionally, rehabilitants and informal caregivers expect to be taken care of during their stay at the rehabilitation ward, and they do not expect to have to perform daily tasks themselves.
		Cognitive aspects	Often, rehabilitants suffer from cognitive problems or delirium. Besides already existing cognitive problems, cognition may decline as a result of the life event or diagnosis for which they are receiving rehabilitation. Neuropsychiatric symptoms such as depression or disrupted stimulus processing occur as a result of a neurological condition. It is important to be aware of these symptoms, as they can affect the rehabilitation process, but also acknowledge that this is an often underexposed and, as a result, under addressed aspect during rehabilitation. Adapting rehabilitation to the needs and learning style of the rehabilitant is important, and professionals must be aware that information and exercises must be offered in different ways.
		Resilience	Rehabilitants’ resilience is often low, especially at the beginning of the rehabilitation process. Rehabilitants and informal caregivers need to understand and learn to deal with this. For balance, it is important to create rest moments for rehabilitants, and therapies must be spread out over the whole week, not just provided during working hours. Participants acknowledge this but have difficulty determining how much rest a rehabilitant needs and how to best provide it.
		Self-reliance	Although all participants consider it important that rehabilitants have self-management abilities and take control of their rehabilitation, not all rehabilitants are able to do this from the start. To be able to take control, rehabilitants must know what the possibilities are and have the opportunity to practice on their own as well as carry out their own planning. Rehabilitants’ motivation can be improved if they know what is expected of them and what they are working for.
Goals	Individual goals are needed for the rehabilitation process in a CRE. Some rehabilitants need guidance in setting their goals. There is a desire for an appropriate set of measurement instruments for GR.	Goal setting	Shared goals for rehabilitation (rehabilitants, informal caregivers and professionals) are important. Not all rehabilitants are able to express their goals at the start, but with support from informal caregivers, relevant goals can usually be defined. Sometimes, their goals are unrealistic for their level of functioning. Goal setting in smaller steps, with good guidance and communication by the professionals and tailored to the rehabilitants’ needs, will improve the chances that rehabilitants will achieve their goals. One of the main goals of rehabilitants is to work on self-reliance and independence to practice what they need to be able to go home. In addition, professionals should be aware of possible cultural differences in the importance of goals.
		Learning new skills	There is a discussion on teaching rehabilitants new skills. Society increasingly demands digital skills. Although the participants think inpatient rehabilitation is a good moment to learn these new skills, they also admit that not all rehabilitants are willing to learn these skills and rarely succeed in reaching a higher level of independence than before.
		Observation and measuring	A wide range of measurement instruments indicate the level of function of a rehabilitant, although not many instruments are validated in the population in GR. Therefore, functional observation (live or by recording) is still often used. Participants long for a set of measurement instruments appropriate for the population, which can be used to motivate and inform the rehabilitant about their progress.
Exercise	Exercise intensity in a CRE is as high as possible. This can be achieved by integrating task-oriented exercises, patient-regulated exercises and group training into the daily structure.	Exercise intensity	Exercise intensity should be as high as possible on all days of the week, based on the rehabilitant’s ability. Currently, this intensity is often not high enough. Exercise intensity comprises all activities as part of the rehabilitation. Rehabilitants, informal caregivers and staff should be aware that it is not only the moments with a therapist that are important for the rehabilitation; they need to integrate training into their daily routine.
		Task-oriented exercises	Although you must sometimes start at the level of body functions, therapy in a rehabilitation setting aims at the participation level. Task-oriented exercise is in line with this. For example, tasks can be practiced in activities of daily living, at mealtimes and in hobbies. All staff must have the attitude and the time to stimulate rehabilitants to practice meaningful tasks, which are tailored to their home situation, throughout the day.
		Patient-regulated exercises	Patient-regulated exercises can increase exercise intensity and stimulate the rehabilitants’ independence and self-management during rehabilitation. Homework exercises can increase the amount of patient-regulated exercise and can affect how rehabilitants continue to perform exercises after discharge. To stimulate patient-regulated exercise, 24/7 access to training facilities is desirable and informal caregivers and staff should stimulate rehabilitants to practice, although independent exercise is often at a lower intensity than supervised training.
		Group training	Group training can be an effective way to increase exercise intensity, but it should be compatible with the goals of the rehabilitant. Training in groups stimulates contact with and learning from others, prevents loneliness, and stimulates rehabilitation. Therefore, it is important that staff members stimulate a positive group process on a rehabilitation ward.
Daily schedule	Within a CRE, the entire day the team needs to be focused on rehabilitation and activities. Exercise is adapted to the pace of the rehabilitant.	Daytime activities	A daily schedule that challenges rehabilitants to take initiative and increase their exercise intensity is desired. When rehabilitants are too passive between therapy moments, this can sometimes lead to cognitive decline. Recreational activities not focused on the rehabilitation goals can keep the rehabilitants motivated, for example, activities with a game element. It is recommended to allow visitors/guests, other than the informal caregivers (who can assist during rehabilitation), only during predefined visiting times.
		Planning	Some rehabilitation wards work with a day planning for the rehabilitants in which all therapies are planned. Some rehabilitants and informal caregivers appreciate this planning, but it often causes problems. Because of external factors, such as hospital visits, the planning needs to be quite flexible. Working without a therapy plan enables responding to the rhythm of the rehabilitant and promotes interdisciplinary cooperation. In addition, it is desirable to have walk-in moments for the therapy so that rehabilitants can take control of when they practice, if they are able to.
Involving client system	Good communication is necessary to involve informal caregivers in the rehabilitation process. They can help in the rehabilitation process, but they should be prevented for overburdening.	Informal caregiver participation	The informal caregivers and their abilities and perceived burden partly determine whether a rehabilitant can go home. They can provide information about the rehabilitant’s previous level of functioning and the goals for the rehabilitation. Although informal caregivers can be seen as fellow practitioners, who motivate and help during the rehabilitation process, staff must prevent overburdening them. Attention must also be paid to bereavement and the informal caregiver’s need for information. Participants would like to see informal caregivers perform tasks in supporting the rehabilitant during rehabilitation similar to what they will be doing at home.
		Communication	Communication is a key aspect in involving informal caregivers. Rehabilitants and informal caregivers need to be informed about the principles of a CRE so that they know how important it is to practice during daily activities and which extra exercises they can perform during the rehabilitation process. This information must be presented in a way suitable for the rehabilitant and informal caregiver. It is also important to give information about the disease for which they are undergoing rehabilitation and about the new skills they must learn.
Nutrition	Nutritional status partly determines the rehabilitation capability, therefore a balanced diet is necessary.		A rehabilitant’s nutritional status partly determines their rehabilitation capability. Rehabilitants are not always aware of this relationship and do not consume enough protein-rich foods. It is important to realize a balanced diet with products that are as common as possible, so rehabilitants will be able to continue the diet at home. Pleasant mealtimes stimulate good intake, and joint meals are therefore seen as standard. It is important to pay attention to the energy levels of a rehabilitant. Intake or swallowing can be negatively affected if a rehabilitant is too tired.
Technology	Technology develops very fast and contributes to safe and challenging rehabilitation.	Domotics	Domotics, e.g., systems to automatically measure body functions or fall signaling, can help to offer security, privacy, and night rest to rehabilitants and can also be time-saving for professionals. An important condition is that privacy is guaranteed and that the security of the system can also be guaranteed at home.
		eHealth	Although the use of eHealth is currently limited, it can be useful in the future as a supplement to exercising, monitoring, safety, and feedback options. Nowadays, many applications are not yet suitable for the target group or are not always applicable during functional activities. In addition, eHealth is developing very fast, making it difficult for healthcare professionals to keep abreast of all possibilities.

**Table 3 jcm-12-01231-t003:** Themes involving organizational aspects.

Main Theme	Brief Description	Sub-Theme	Description
Environmental aspects	The environment on a rehabilitation ward is safe and invites rehabilitants to practice as much as possible.	Building aspects	The environment on a rehabilitation ward should resemble the domestic situation as much as possible so that rehabilitants feel stimulated, free, and able to practice meaningful, functional tasks. There should be therapy rooms on the ward for interdisciplinary team dynamics and the participation of informal caregivers. It is advisable to have the possibility to screen off part of the room for privacy. Walking distances to the bedrooms should be considerable, and there should be handrails, chairs, and exercise facilities in the corridors and a possibility to go outside. For different levels of stimulus processing, there must be variation in rooms with more and less stimulus. Opinions differ as regards the desire for single or multi-bedrooms. Single bedrooms offer privacy and a quiet environment, whereas multi-bedrooms offer contact with other rehabilitants. A sliding wall can offer a solution. The bedroom should be furnished in a way that the rehabilitant is stimulated to get out of bed.
		Ambiance	The ambiance should enthuse rehabilitants and make them feel safe enough to work on their recovery. Most participants think a cozy, homely ambiance is important to stimulate rehabilitants to practice, have contact with fellow rehabilitants, and encourage each other to practice. Relaxing activities should be scheduled in addition to therapy moments.
Staff aspects	All team members work in an interdisciplinary way and stimulate rehabilitants to practice throughout the day.	Team mix	The team should be sufficiently ‘mixed’ in terms of rehabilitation skills and experience. Recommended professionals in the rehabilitation team are: nurses, physical therapists, occupational therapists, psychologists, social workers, case managers, dieticians, speech and language therapists, physicians (elderly care or rehabilitation), and volunteers. Some participants think the nurse needs a name that better reflects the role of a therapeutic rather than caring nurse, for example, “rehabilitation coach”. Regardless of the name, the nurse must be seen as a therapeutic team member.
		Team dynamics	In an interdisciplinary team, each discipline has expertise in a particular area, but team members can look beyond the boundaries of their own field. All disciplines are equal, and there is no (in)formal hierarchy. Working in smaller teams, taking courses together, and therapists working directly on the ward are ways to improve interdisciplinary dynamics. The rehabilitant and their informal caregiver must also be part of the team. Multidisciplinary consultation in the presence of the rehabilitant is preferred and is often used to coordinate rehabilitation goals.
		Attitude of staff	All employees should have an empathetic, motivating attitude in order to involve informal caregivers and stimulate rehabilitants to practice throughout the day. They, therefore, need to be able to see training opportunities in daily activities. The approach of the team is coordinated, so rehabilitants always know what to expect. Staff members ideally choose to work in the field of rehabilitation, are flexible, can set priorities, have an interdisciplinary mindset, and are stress resistant.
		Training requirements	Medical guidelines are not always suitable for geriatric rehabilitants. Staff must be able to deal with this by using evidence-based practice principles, building sufficient experience and having additional training in geriatrics.
Organizational aspects	Implementing a CRE requires a shared vision on rehabilitation, and a project group to supervise the process. Even though internationally the organizational aspects differ, the concept of CRE is suitable to get the most out of rehabilitation.	Vision	It is important that all professionals (including management) have a shared vision on geriatric rehabilitation and make informed decisions.
		Administration	Participants experience that too much time is spent on administrative tasks, partly due to incompatible systems and regulations—time that could be spent on the rehabilitants and their rehabilitation.
		Regulations and funding	Participants feel that the rehabilitation system is driven by the way it is funded, which differs internationally. Optimal rehabilitation cannot always be offered due to insufficient reimbursement. Unfavorable decisions are sometimes unavoidable within the therapies. Participants experience a negative effect of regulations regarding privacy and liability in the rehabilitation process.
		Safety	Participants think rehabilitants and informal caregivers need to be safe to practice. The approach of the professionals and the design of the building may affect this safety positively or negatively. Despite some international differences, participants agree that pushing the boundaries, taking calculated fall risk, and using technical innovations to prevent risks will improve rehabilitation.
		Different settings	There are international differences in the setting in which rehabilitation for older persons is offered, and whether it is separate from rehabilitation for younger adults. However, these differences are secondary: the concept of a rehabilitation environment must start at the hospital ward and should continue after discharge in a slightly modified form.
		Naming	Sometimes, the word rehabilitation “hotel” is used for a rehabilitation ward, which may create expectations of being pampered instead of there being hard rehabilitation work perform. Using the word “patient” emphasizes being ill. Using the word “person” or “rehabilitant” stimulates looking at a person’s abilities.
		Implementation	Implementing a CRE requires a balanced team, and all team members must agree on the need for the implementation of a CRE. A project group or initiator should supervise the implementation of themes within CRE, work on time management and keep everyone enthusiastic. It takes a lot of time for a new method to become fully embedded in the daily routine.
Factors outside the ward	The discharge process must be well prepared and supervised. Home visits allow rehabilitants to practice meaningful tasks in their own environment and be prepared for discharge.	Outpatient rehabilitation	It is important to visit the home environment during inpatient rehabilitation because the situation at home can be different from the rehabilitation ward. It is best for rehabilitants to practice meaningful tasks in their own environment. However, some participants think rehabilitating in a good CRE can have added value.
		Discharge process	It can be beneficial to guide the transition home by continuing the rehabilitation process by the same professionals in outpatient rehabilitation. Rehabilitants and informal caregivers sometimes think they are not ready to be discharged, while professionals think they can manage at home. It is important to keep communicating about the discharge process. Additionally, longer rehabilitation can sometimes be beneficial to increase a rehabilitants’ independence, which subsequently leads to lower healthcare costs in the long term.

## Data Availability

The data presented in this study are available on request from the corresponding author. The data are not publicly available due to privacy.

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
