# Peer review of "A Qualitative Study Exploring Professional Perspectives of a Challenging Rehabilitation Environment for Geriatric Rehabilitation"

_jcm, 2023, doi:10.3390/jcm12031231_

Round 1

Reviewer 1 Report

I appreciate your time and effort in performing a qualitative study regarding the components required for modeling a CRE for geriatric rehab. Geriatric rehab is a growing component of medicine within Western society given the aging of the population and frequency of hospitalizations, with attendant functional and cognitive decline.

1.       Methods were clearly described. I appreciate your efforts to include individuals with significant knowledge of CRE from a variety of disciplines and agree that dividing them into different groups to avoid possible hierarchal impacts on their responses was important. Including sessions at EuGMS was important to capture input from rehab experts outside of the Netherlands. The focus groups and workshops are well described. Inclusion of the topic list in Appendix 1 adds to the impact of the manuscript.

2.       The data analysis is clearly presented. I applaud you on reaching data saturation.

3.       Results are well presented. The number of participants is quite reasonable and includes a good variety of occupations. I do note that all are from Europe dictating recording the limited scope of the nationalities of participants is noted as a possible limitation. Future study would benefit from including rehabilitation specialists from other parts of the world. Tables 2 and 3 are excellent. I applaud you for clearly laying out the main and sub-themes and including a description. This will be very helpful for individuals working to create or improve geriatric rehabilitation within a CRE.

4.       The discussion was well organized and comprehensive. Appreciate that you highlight the need to tailor rehab to the resilience, goals, and cognition of the participant. This aligns with patient-centered care and the Age-Friendly movement in the US. Agree that further study is needed regarding the benefits for a structured CRE versus home rehab where individuals can work within their own environment.

5.       The English should be reviewed. Some grammatical corrections are needed to ensure clarity and enhance readability.

Author Response

Dear Reviewer 1,

Thank you for reviewing our manuscript and your nice words.

We are very grateful that you are positive about our study, and the way how we described the study results in our manuscript.
In line with your advice regarding the readability and clarity of our manuscript, we have used the language editing service from MDPI.

Kind regards,

Lian Tijsen MSc.
dr. Els Derksen
prof. dr. Wilco Achterberg
dr. Bianca Buijck

Reviewer 2 Report

It was with great interest I agreed to review this manuscript “A qualitative study exploring professional perspectives of a challenging rehabilitation environment for geriatric rehabilitation” as this was a new concept to me. The authors state that there is a trend towards formalization of the rehabilitation process for older rehabilitants in a Challenging Rehabilitation Environment (CRE). It appears that this applies to the country in which the study was conducted. This does not necessarily limit the interest of an international audience if the concept is well defined, theoretically underpinned, delineated to similar concepts and if available supported by empirical studies. However, the concept is poorly defined and the theoretical underpinnings are lacking. The lack of conceptual clarity have repercussions on the whole manuscript. I regret to say that after reading the manuscript several times I still don’t understand what the concept CRE entails.

The concept challenging rehabilitation environment according to the authors “involves the comprehensive organization of care and support by the rehabilitation team as well as the environment in which the rehabilitation takes place “ So how does CRE differs from the concept rehabilitation? What does challenging entail in this context, and for whom? Environment is a wide concept how is it defined in this context, place, space, or?

The rationale is unclear.

Furthermore the aims of the study are varying e.g., This study aims to explore the opinions regarding CRE of healthcare professionals. We explore which aspects professionals consider necessary for modelling an effective CRE. To explore the perspectives of professionals on the concept of CRE. Somewhat similar but different aims.

The authors should be commended for the recruitment of a large group of participants. The method how the data were collected are clear as well as the analyses. Yet the results are very descriptive and lack the depth you would expect in a qualitative study in which the first topic addressed in the topic list is the meaning of the concept.

How data are categorized by the authors can be easily followed but how the different themes fit into the categories rehabilitation process versus organizational aspects can be discussed.  E.g., the organizational aspect “outside the ward” contains the discharge process -  could be argued that it is also a part of the rehabilitation process. The presentation in detailed table and the similar text in the running text is somewhat redundant.  Some themes only appear in the running text e.g, involving client system, versus involving informal care system in the table. 

In the discussion the authors make several statements that are not supported by the results. e.g., To offer effective rehabilitation, all elements of CRE should be applied. How did the authors arrive at this statement? Is this what was conveyed by the participants in the focus groups? If so, this should be explicitly stated. 

It is unclear if new knowledge has been generated through the study.

Author Response

Dear reviewer 2,

Thank you for your time and effort in reviewing our manuscript, and for the valuable comments. Below we will answer your comments point by point.

  • The authors state that there is a trend towards formalization of the rehabilitation process for older rehabilitants in a Challenging Rehabilitation Environment (CRE). It appears that this applies to the country in which the study was conducted.

CRE indeed originated in the Netherlands. Therefore, in the introduction in line 58, we write that this concept is the trend in the Netherlands. In line 72 we mention that internationally, CRE is a relatively new concept. More specific, in the UK and in Greece, there is a growing interest in the principles of CRE.

In the Netherlands, there is a trend towards having the rehabilitation process take place within the context of a Challenging Rehabilitation Environment (CRE). A CRE is a widely accepted practice based innovation in the Netherlands.

Although  internationally there is a growing interest in the principles of this relatively new concept, there is no official scientific based definition of CRE.

  • the concept is poorly defined and the theoretical underpinnings are lacking. The lack of conceptual clarity have repercussions on the whole manuscript.

CRE is a practice based innovation, which is widely accepted in the Netherlands, and there is a growing interest from other countries. However, there is no theoretical underpinning of the concept. In the CREATE study we make the first steps towards that theoretical underpinning.

In the manuscript we rephrased and added sentences to clarify.  

In the Netherlands, there is a trend towards having the rehabilitation process take place within the context of a Challenging Rehabilitation Environment (CRE). A CRE is a widely accepted practice based innovation in the Netherlands. It is a unique concept which is positioned on the rehabilitation ward, covering all rehabilitation-aspects that possibly influence rehabilitation outcomes. The concept involves the comprehensive organization of care and support by the rehabilitation team as well as the environment in which the rehabilitation takes place. In comparison to regular rehabilitation (with mostly mono-disciplinary goals and interventions), in a CRE the rehabilitation interventions  are integrated in all aspects of the day and daily life, and the rehabilitation process is offered in an interprofessional way, with team & rehabilitant goals and interventions.
A review conducted by the authors initially identified seven main components for modelling CRE: 1) therapy time; 2) group training; 3) patient-regulated exercise; 4) family participation; 5) task-oriented training; 6) enriched environment; and 7) team dynamics.[9]
Although  internationally there is a growing interest in the principles of this relatively new concept, there is no official scientific based definition of CRE. This results in considerable differences between rehabilitation wards in the interpretation of rehabilitation in general and execution of CRE specifically.

  • So how does CRE differs from the concept rehabilitation? What does challenging entail in this context, and for whom? Environment is a wide concept how is it defined in this context, place, space, or?

There are indeed similarities between regular rehabilitation and CRE. The unique aspect of CRE is that rehabilitation aspects are integrated in all aspects of the day and daily life and the rehabilitation process is offered in an interprofessional way with shared goals and interventions.
This is for instance shown in the interior design of the rehabilitation institution. In a CRE there are possibilities to practice in every possible corridor and room of the ward and institution. In a typical/general/regular rehabilitation institution, the therapies mostly take place in the therapy room.

The interprofessional way of working, is shown in the fact that all professionals are responsible for achieving the goals of a rehabilitant. Therefore, the rehabilitation goals are interdisciplinary formulated on the level of participation, instead of mono-disciplinary formulated on the level of a specific rehabilitation outcome.

In the manuscript in line 60 we have added:

It is a unique concept which is positioned on the rehabilitation ward, covering all rehabilitation-aspects that possibly influence rehabilitation outcomes. The concept involves the comprehensive organization of care and support by the rehabilitation team as well as the environment in which the rehabilitation takes place.  In comparison to regular rehabilitation (with mostly mono-disciplinary goals and interventions), in a CRE the rehabilitation interventions  are integrated in all aspects of the day and daily life, and the rehabilitation process is offered in an interprofessional way, with team & rehabilitant goals and interventions.

  • The rationale is unclear.

We hope that the answers to the 3 above questions provide more clarity with regard to the rationale.

  • Furthermore the aims of the study are varying e.g., This study aims to explore the opinions regarding CRE of healthcare professionals. We explore which aspects professionals consider necessary for modelling an effective CRE. To explore the perspectives of professionals on the concept of CRE. Somewhat similar but different aims.

Thank you for noticing this inconsistency. We have changed the text in line 26 and 78-79 to use consistent words:

This study aims to explore the perspectives regarding CRE of healthcare professionals through a qualitative study.

In this part of the CREATE study we explore the perspectives of professionals regarding CRE.

  • The authors should be commended for the recruitment of a large group of participants. The method how the data were collected are clear as well as the analyses.

Thank you for these nice words.

  • Yet the results are very descriptive and lack the depth you would expect in a qualitative study in which the first topic addressed in the topic list is the meaning of the concept.

The results that are described in the manuscript are the results of the data of our focus groups and workshops. These results give a description of the concept of CRE.
To create an in-depth conceptualization, these results need to be combined with the perspectives of rehabilitants and informal caregivers and with evidence from literature.
The overall aim of the CREATE study is to create this in-depth conceptualization, therefore we will address this aspect in further research, which may lead to the formulation of a theory regarding CRE.
As mentioned in the conclusion, this is the first step towards the conceptualization of CRE.

  • How data are categorized by the authors can be easily followed but how the different themes fit into the categories rehabilitation process versus organizational aspects can be discussed.  E.g., the organizational aspect “outside the ward” contains the discharge process -  could be argued that it is also a part of the rehabilitation process.

Indeed the main theme “Factors outside the ward” have been thoroughly discussed by the research group, and could also be placed in the category regarding the rehabilitation processes. As this theme contains organizational aspects regarding how to organize the discharge process, we have decided to categorize this theme in the category organizational aspects.

  • The presentation in detailed table and the similar text in the running text is somewhat redundant.  Some themes only appear in the running text e.g, involving client system, versus involving informal care system in the table. 

The manuscript should be readable without reading the tables and the tables give some extra in-depth information about the themes. Therefore, this can feel a bit redundant.

Thank you for noticing this inconsistency in the naming of the themes. We have changed informal care system to client system in line 252 and in the table.

  • In the discussion the authors make several statements that are not supported by the results. e.g., To offer effective rehabilitation, all elements of CRE should be applied. How did the authors arrive at this statement? Is this what was conveyed by the participants in the focus groups? If so, this should be explicitly stated. 

The results described are based on the perspectives of 180 professionals working in rehabilitation. The themes in this manuscript are formulated based on their perspectives. This has lead to the conclusion that these themes are necessary for an effective CRE.

We have changed the last sentence of the discussion to:

Consequently, effectivity and efficiency of CRE need to be studied with validated tools yet to be developed.

In our ongoing research we aim to develop those tools.

  • It is unclear if new knowledge has been generated through the study.

Prior to the CREATE study, there was ambiguity about the interpretation of CRE. Even though, the individual aspects of CRE may not be new, the coherence of these aspects is.

We hope that the changes we made are satisfactory. We are looking forward to your answer regarding our revised submission. We would be pleased to respond to any further questions and comments that you may have.

Kind regards,

Lian Tijsen MSc.
dr. Els Derksen
prof. Dr. Wilco Achterberg
dr. Bianca Buijck